# Long-Range and Directional Allostery of Actin Filaments Plays Important Roles in Various Cellular Activities

**DOI:** 10.3390/ijms21093209

**Published:** 2020-05-01

**Authors:** Kiyotaka Tokuraku, Masahiro Kuragano, Taro Q. P. Uyeda

**Affiliations:** 1Department of Applied Sciences, Muroran Institute of Technology, Muroran, Hokkaido 050-8585, Japan; gano@mmm.muroran-it.ac.jp; 2Department of Physics, Faculty of Science and Engineering, Waseda University, Tokyo 169-8555, Japan; t-uyeda@waseda.jp

**Keywords:** long-range allostery, actin, actin-binding protein, cofilin, cooperativity, drebrin, filamin, fimbrin, gelsolin, myosin, tropomyosin

## Abstract

A wide variety of uniquely localized actin-binding proteins (ABPs) are involved in various cellular activities, such as cytokinesis, migration, adhesion, morphogenesis, and intracellular transport. In a micrometer-scale space such as the inside of cells, protein molecules diffuse throughout the cell interior within seconds. In this condition, how can ABPs selectively bind to particular actin filaments when there is an abundance of actin filaments in the cytoplasm? In recent years, several ABPs have been reported to induce cooperative conformational changes to actin filaments allowing structural changes to propagate along the filament cables uni- or bidirectionally, thereby regulating the subsequent binding of ABPs. Such propagation of ABP-induced cooperative conformational changes in actin filaments may be advantageous for the elaborate regulation of cellular activities driven by actin-based machineries in the intracellular space, which is dominated by diffusion. In this review, we focus on long-range allosteric regulation driven by cooperative conformational changes of actin filaments that are evoked by binding of ABPs, and discuss roles of allostery of actin filaments in narrow intracellular spaces.

## 1. Introduction

Actin filaments play many important roles in eukaryotic cells by interacting with various actin-binding proteins (ABPs). To realize these activities, the localization of ABPs is spatially and temporally regulated in cells. For example, myosin II is localized only in the posterior region in a migrating *Dictyostelium* cell, even though the anterior region is rich in actin filaments [1,2]. On the other hand, cofilin is localized to lamellipodia in the anterior region, and recycles actin for continuous polymerization at the leading edge [3]. This different localization of ABPs is one driving force for cell migration. Nonmigratory cells have other types of actin-based structures involved in cell-cell and cell-substrate adhesion such as adherens junctions, tight junctions and focal adhesions, as well as more specialized structures such as microvilli and stereocilia. Again, each of those structures depends on spatially-regulated interactions of actin filaments with specific sets of ABPs. Although the regulation of ABP localization is generally explained by local biochemical signaling such as phosphorylation and changes in the concentration of signaling molecules, not all aspects of ABP regulation can be explained by local biochemical signaling [4], and the overall picture is still unclear.

Recent developments in structural biology have revealed dynamic structural polymorphism in pure actin filaments [5,6]. Furthermore, various ABPs are now known to induce cooperative conformational changes of actin filaments [7], thereby controlling the binding of ABPs to actin protomers not in direct contact with the bound ABP in vitro, which we call long-range allostery. In addition to local biochemical signals, long-range allostery in actin filaments may be involved in controlling the localization of ABPs in micrometer-scale intracellular spaces. In this review, we focus on the spatial and temporal allosteric regulation of ABP localization by cooperative conformational changes of actin filament cables.

## 2. Cooperative Binding of ABPs

For decades, cooperative binding of various ABPs to actin filaments, accompanied by cooperative conformational changes in filaments, has been reported. Such ABPs include myosin II [8,9,10,11,12,13,14,15,16,17,18,19,20,21,22,23], tropomyosin [24,25,26,27], cofilin [4,28,29,30,31,32,33,34,35,36,37,38,39,40,41], drebrin [42,43], fimbrin [44,45,46], α-actinin [47,48], filamin [49], α-catenin [50], gelsolin [51,52] and formin [53]. In recent years, the interaction between different ABPs via conformational changes of actin filaments, including a mutually exclusive interaction, has also been reported. In this section, we will list ABPs that have been reported to bind cooperatively with actin filaments. They are classified by one of three binding styles—side-binding, cross-linking, and end-binding.

### 2.1. Side-Binding ABPs

#### 2.1.1. Myosin II and V

Myosin [54] is a family of motor proteins that convert chemical energy of ATP into motion energy. There are a large number of reports of cooperative conformational changes within actin filaments induced by the binding of myosin [8,9]. Nearly fifty years ago, Oosawa et al. first reported the cooperative interaction between heavy meromyosin (HMM) of myosin II and actin filaments, observing that the binding of HMM increased the fluorescence intensity of labeled actin, with the effect being saturated in the presence of a 1:20 molar ratio of HMM to actin protomers [10]. Miki et al. also reported that the maximal change in fluorescence in actin filaments occurred when only one HMM molecule was bound per 50 actin protomers or when one subfragment 1 (S1) molecule was bound per 25 actin protomers [11]. There are many similar reports on the cooperative interaction between myosin II and actin filaments [12,13,14,15]. Cooperative binding of myosin II and actin filaments was observed by electron microscopy [16,17,18,19,20]. For, example, Orlova and Egelman reported very large cooperativity in the binding of HMM to actin filaments with Ca^2+^ bound at the high-affinity metal binding site [20]. Cooperative binding was also reported by fluorescence microscopic observation using GFP-fused HMM (HMM-GFP) derived from *Dictyostelium* myosin II [21]. When HMM-GFP was mixed with actin filaments with either Ca^2+^ or Mg^2+^ at the high-affinity metal binding site, HMM-GFP formed clusters along both forms of actin filaments. The density of HMM molecules in the cluster along Ca^2+^-actin filaments was higher than that along Mg^2+^-actin filaments [21]. The weak cooperative binding to Mg^2+^-actin filaments required low, submicromolar concentrations of ATP [21]. The growth of HMM-GFP clusters along Mg^2+^-actin filaments, which is the physiological form [22], was unidirectional, as determined by real-time fluorescence microscopy [23]. Cooperative cluster formation was observed along Mg^2+^-actin filaments that were loosely immobilized on positively charged lipid bilayers but not when tightly immobilized by biotin-avidin linkage, suggesting that the myosin-induced cooperative conformational changes in actin filaments involves a change in the helical pitch [23]. Kozuka et al. examined the dynamic polymorphism of actin filaments by using single molecule intramolecular Förester resonance energy transfer (FRET) imaging, noting that actin protomer switches between low- and high-FRET efficiency states [6]. The high-FRET efficiency state is favored when actin filaments interact with myosin V in the presence of ATP [6], suggesting that myosin V can also induce a cooperative conformational change in actin protomers.

#### 2.1.2. Tropomyosin

Tropomyosin [55] is a regulatory protein found in muscle and non-muscle cells. In striated muscle, this protein functions in association with actin filaments and troponin to confer calcium sensitivity to the actomyosin ATPase [56,57]. Cooperative effects between tropomyosin and actin filaments have recently been reported. For example, proteolytic modification within the DNase-binding loop of actin increased the rate of subunit exchange along actin filaments and tropomyosin binding almost completely suppressed the increase in subunit exchange [25]. The effect was cooperative, with half-maximal inhibition observed at about a 1:50 molar ratio of tropomyosin:actin [25]. In mammals, >40 tropomyosin isoforms can be generated through alternative splicing from four *tropomyosin* genes. Interestingly, the binding of these isoforms is segregated to different actin filament structures in nonmuscle cells, presumably because of cooperative interactions [27].

#### 2.1.3. Cofilin

Cofilin and the closely related actin-depolymerizing factor (ADF) control actin polymerization and depolymerization in a pH-sensitive manner, and inhibit interaction of actin filaments with myosin and tropomyosin [58,59]. Cofilin also binds to actin filaments cooperatively [28,29,30,31] and forms tight clusters [4,32,33,34,35,36,37,38]. Severing of actin filaments by cofilin often occurs at or near the boundaries between cofilin clusters and the bare zones, presumably because of structural discontinuities at those sites [32,36,38]. Actin filaments bound with cofilin clusters were supertwisted by 25% [32,33,34,35], and high-speed atomic force microscopy (HS-AFM) revealed that the supertwisted conformation propagates unidirectionally to the neighboring bare zone on the pointed-end side of the cluster [32]. On the other hand, fluorescence microscopy revealed that cofilin clusters grow bidirectionally [38]. The cause of this discrepancy is currently unknown, but it may be due to the difference in resolution between AFM and fluorescence microscopy or to differences in the method of immobilizing actin filaments to the substrate. The cooperative binding of cofilin to actin filaments has been analyzed using actin mutants as well. For example, actin of the G146V mutant fails to bind cofilin. Intriguingly, binding of cofilin to copolymers of wild type and mutant actins was strongly inhibited even when the ratio of wild type actin and G146V actin in copolymers was 9:1, suggesting that the cooperative conformational change propagates about 10 molecules of actin protomers [39]. K336I mutant actin also cooperatively inhibited binding of cofilin to wild type actin in a copolymer [41]. Moreover, HS-AFM demonstrated that approximately a half helical pitch, which contains 14 actin protomers, of the bare zone on the pointed end side of a cofilin cluster is supertwisted [32]. Recent cryoelectron microscopic analysis suggested that only one or two actin protomers adjacent to cofilin clusters show maximum binding cooperativity [40], and again, the source of discrepancy between this and the HS-AFM observation is currently unknown. On the other hand, differential scanning calorimetry demonstrated that one molecule of cofilin affects the structure of about 100 actin protomers [60]. These results suggest that cofilin induces a wide range (~100 protomers) of structural changes in actin filaments, which may be a different type of cooperative conformational change from that which occurs in a narrower range (1–10 protomers) involving changes in the helical pitch.

#### 2.1.4. Drebrin

Drebrin [61] is an actin-binding protein in the brain that regulates synaptic plasticity of neuronal dendrites [62]. Drebrin A induces cooperative change in the helical structure of actin filaments and cooperatively binds to filaments [42,43]. AFM analysis showed that drebrin A-induced structural and mechanical remodeling in actin filaments involves significant changes in helical twisting and filament stiffness [43]. Drebrin A N-terminal fragment (1-300) containing an actin-binding domain (ABD) and an ADF homology domain forms clusters (average cluster size was ~107 nm or ~2.6 actin filament helical repeats) along actin filaments [42].

### 2.2. Cross-Linking ABPs

#### 2.2.1. Fimbrin

Fimbrin [63], referred to as plastin in humans, belongs to a superfamily of actin cross-linking proteins that share calponin homology (CH) domains. In addition to fimbrin, the actin cross-linking proteins with CH domains include α-actinin, filamin, spectrin, dystrophin and ABP-120 [64,65,66,67]. The N-terminal CH domains of fimbrin induced a conformational change in actin filaments [44,45,46]. Fimbrin inhibits tropomyosin binding to actin filaments [46]. It would be interesting to test if this inhibition is due to simple competition of the binding sites on actin protomers, or due to the fimbrin-induced cooperative conformational changes in actin filaments.

#### 2.2.2. α-Actinin

α-Actinin [68] forms a dimer and cross-links actin filament cables to form parallel bundles. α-Actinin also enhances nucleotide exchange of bound actin filaments, and this activity was maximal when α-actinin was added at a 1:49 molar ratio of α-actinin to actin protomers [47], strongly suggesting the cooperative impact of α-actinin binding to many actin protomers.

#### 2.2.3. Filamin

Filamin [69] is also a dimeric actin filament cross-linking protein [70]. The meshwork of actin filaments cross-linked by filamin is involved in many cellular activities such as cell migration, chemotaxis and mechanosensing [71,72,73,74]. The ABD of filamin selectively binds to actin filaments in the rear of migrating cells and contributes to the posterior localization of filamin [49]. The speed of translocation was much faster than that of the retrograde flow of cortical actin filaments, suggesting that the ABD of filamin diffusing in the cytoplasm recognizes a certain feature of the actin filament structure in the rear of the cell and selectively binds to it [49].

Further detailed studies are needed to obtain direct evidence for long-range allostery involving ABDs of cross-linking ABPs.

### 2.3. End-Binding ABPs

#### 2.3.1. Gelsolin

Gelsolin [75] was discovered as a calcium-dependent regulatory protein that controls cytoplasmic actin gel-sol transformation. Low resolution electron microscopy revealed that the binding of one molecule of gelsolin at the barbed end of an actin filament changes the structure of all actin protomers in a filament, representing a case of extremely long-range cooperative conformational change in actin filaments [51]. The amino-terminal half of gelsolin, G1-3, and full-length gelsolin both quenched the pyrene fluorescence of actin, and the extent of quenching and stoichiometry were identical between G1-3 and gelsolin. In contrast, severing of actin filaments by G1-3 was much less efficient than by full-length gelsolin. Quantitative experiments suggested cooperative interactions in which the binding of two G1-3 molecules in close proximity leads to cooperative severing of the polymer, thus increasing severing efficiency [52].

#### 2.3.2. Formin

Formin dimer [76] is a ubiquitous actin filament nucleator that progressively elongates filaments by incorporating actin monomers complexed with profilin [77]. One formin dimer at the barbed end can affect the dynamic properties of the entire filament. Analyses of the results obtained at various formin/actin concentration ratios indicated that at least 160 actin protomers are affected by the binding of a single formin dimer to the barbed end of a filament [53].

## 3. Allosteric Interaction between ABPs and Actin Filaments Involved in the Formation and Function Mechanism of the Actin-Based Machinery

The actin-based motile machineries, which include stress fibers, lamellipodia, filopodia, invadopodia, muscle, and contractile rings, are formed by the interaction between actin filaments and specific ABPs (Figure 1). As mentioned above, the interaction of ABPs with actin filaments induces structural changes of filaments, which regulates the interaction between ABPs. Thus, allosteric interactions involving multiple ABPs may control the formation and function of actin-based machineries, but research in this area has only just begun. This section describes the formation and function mechanisms and characteristics of these machineries, focusing on the stress fibers, lamellipodia, and filopodia (Figure 1), which have been studied to some extent already.

### 3.1. Stress Fibers

Stress fibers are contractile force-generating bundled structures mainly composed of actin filaments and myosin II filaments [79], and are found in nonmuscle cells such as fibroblasts, endothelial cells, myofibroblasts and epithelial cells. The coordination of stress fibers and cell adhesion controls cell migration and cell morphogenesis. Stress fibers contain proteins with various functions, such as actin crosslinking proteins and kinases, in addition to actin filaments and myosin II [80]. The formation and contraction of stress fibers are controlled by the coordinated function of these proteins. The cooperative interaction between actin filaments and ABPs presumably plays a role in the correct assembly of stress fibers. For example, if the actin-depolymerizing protein, cofilin, enters a stress fiber, actin filaments constituting the stress fiber will be severed and its structure will be disrupted.

Cofilin binding supertwists the helix of actin filaments by about 25%, and this conformational change is propagated to a neighboring bare zone [32,33,34,35]. On the other hand, the helix slightly untwists when the myosin motor domain binds in the rigor state [81,82]. This allosteric regulation by cooperative conformational changes of actin filaments may contribute to mutually exclusive binding of cofilin and myosin II [4] and/or tropomyosin [26] to actin filaments. However, the presence of ATP was essential for the mutually exclusive binding of myosin II and cofilin [4], suggesting that the structural change of actin filaments induced by the active myosin heads is different from that induced by rigor crossbridges, and is important for long-range allostery. Additionally, cofilin cannot bind to tensed actin filaments in vivo and in vitro [83]. Since interactions with myosin II in the presence of ATP generates tension, myosin II can generate long-range allostery in actin filaments through two mechanisms, i.e., a direct structural effect and the generation of tension, which would synergistically inhibit the binding of cofilin to actively contracting stress fibers. This would partially explain why stress fibers are disassembled when myosin II is pharmacologically inhibited [84,85] or when the deformable substrate to which cells adhere is relaxed [80,83,86].

Stress fibers also include tropomyosin [87] and α-actinin [88], which interact cooperatively with actin filaments [25,47]. Tropomyosin 4 recruits myosin II to stress fiber precursors, and tropomyosin 1, 2/3 and 5NM1/2 stabilize the actin filaments of stress fibers [89]. Although details of these mechanisms are unknown, it is possible that the cooperative interaction between myosin II and tropomyosin 4 or α-actinin is involved. In addition to the mutually exclusive effects between myosin II and cofilin, these positive cooperative effects may control the proper self-assembly of the actin filament-based molecular machineries. Actin structural changes caused by the binding of myosin II, tropomyosin and α-actinin are all transmitted up to about 50 actin protomers ahead, implying that these ABPs induce structural changes of actin protomers using a similar mechanism. It is likely that the cooperative structural changes of actin filaments evoked by the binding of these ABPs and the tension generated by myosin II are involved in the resistance of stress fibers to cofilin (Figure 2), but also to recruit myosin II. This speculation is supported by the observation that the motor domain alone [90] or S1 [91] of the IIB isoform of nonmuscle myosin II binds to stress fibers in normal rat kidney (NRK) cells or fibroblasts, in the absence of any known biochemical regulation. Moreover, the motor domain or S1 of myosin II is localized to tensed, myosin II-containing actin structures, such as contractile rings in dividing cells and the posterior cortex of migrating cells, in Dictyostelium [92]. This latter result is noteworthy in that it suggested that myosin II-induced and/or tension-induced conformational changes in actin filaments are sufficient to recruit myosin II, since Dictyostelium does not have tropomyosin.

In the NRK study by Tang and Ostap, unlike the motor domain of myosin II, the motor domain of myosin I did not localize to stress fibers. Furthermore, the response to tropomyosin binding to actin filaments was different between myosin I and IIB. Interestingly, S1 of the IIA isoform of nonmuscle myosin II did not bind to stress fibers [91]. It is therefore unlikely that the motor domains of all classes of myosin, or even different isoforms within a specific class, induce similar cooperative conformational changes to actin filaments, or recognize a similar structure of actin filaments for cooperative binding. Each isoform or class of myosin appears to be optimized in terms of inducing and/or recognizing cooperative conformational changes in actin filaments.

### 3.2. Lamellipodia

Lamellipodia are broad and flat protrusions along the leading edge of a migrating cell, including neural crest derived cells in several species, keratocytes and macrophages in *Drosophila melanogaster* [93]. They are also composed of actin filaments and various ABPs, such as formin, profilin, the ARP2/3 complex and cofilin [78]. Lamellipodia drive cell migration by the force of actin polymerization at the leading edge. Therefore, it is important that G-actin is recruited to the tip of filaments to maintain continuous polymerization. G-actin is supplied by recycling from old filaments that interact with cofilin [94].

HS-AFM observation revealed that shortening of the helical pitch of actin filaments induced by cofilin binding propagate to the pointed-end side of the cofilin cluster, and that cofilin clusters grow unidirectionally toward the pointed-end of the filament [32]. Severing by cofilin was often observed near the boundaries between cofilin clusters and bare zones or within each cofilin cluster [32]. Cluster growth on the pointed-end side of the initially bound cofilin molecule may accumulate strain within the cluster, resulting in severing of actin filaments at the boundary between the cluster and the bare region and/or inside the cluster. In lamellipodia, the pointed end and barbed end of actin filaments are oriented toward the nucleus and leading edge of the cell, respectively. Formin is bound to the barbed end of actin filaments [78]. Helical rotation of formin decreases the helical twist of an actin filament if there is rotational friction between the formin molecule and the surroundings and between the actin filament and the surroundings. This would confer resistance to cofilin [95]. Cofilin interacts with actin filaments in sections near the pointed end and depolymerizes filaments because structural changes due to formin bound at the barbed end are probably not transmitted to the sections close to the pointed end due to the branch involving the ARP2/3 complex [78,96] (Figure 3). This structural effect, together with the differential distribution of ATP-, ADP•Pi- and ADP-bound protomers along the filament [97,98], which is not discussed in this article, may allow cofilin to efficiently depolymerize actin filaments deeper inside a cell, which are no longer necessary for the generation of force along the leading edge of a lamellipodium.

### 3.3. Filopodia

Filopodia are thin (diameter 0.1-0.3 µm), finger-like projections containing parallel bundles of actin filaments, and function as an antenna for detecting the cell environment [78,99]. The major actin filament-bundling protein in filopodia is fascin [100]. The homologous molecules of fascin have been discovered in a wide range of species, from insects to mammals [101]. The distribution of fascin throughout the actin filament bundles in filopodia [102,103] suggests that fascin can bind to actin filaments independently of formin binding and stabilize filopodia (Figure 4). The crosslinking of actin filaments by fascin presumably suppresses the propagation of structural changes of filaments, including not only untwisting by formin but also supertwisting by cofilin. In fact, Hirakawa et al. reported that cofilin was difficult to cooperatively bind to actin filaments tightly immobilized on a glass surface using biotin-avidin linkages [23]. It is likely that filopodia show resistance to cofilin due to the untwisting of filaments by formin and the crosslinking of filaments by fascin (Figure 4).

It has not been reported whether fascin induces cooperative structural changes of actin filaments, but a small-angle x-ray scattering study revealed that fascin bundles actin into a continuous spectrum of intermediate twist states [104]. Fascin and α-actinin can work in concert to generate enhanced cell stiffness [105], and myosin V selectively moves to the tip of filopodia along the actin filament bundled by fascin [106]. These results imply that there is also an unknown cooperative interaction between fascin and actin filaments.

### 3.4. Other Actin-Based Machineries

Muscle is the most representative and well-studied actin-based machinery. Proteins constituting myofibrils are classified into contractile, regulatory, and structural proteins. Actin and myosin II, which are contractile proteins, polymerize to form thin and thick filaments, respectively, and muscles can contract as they slide on each other [54]. Tropomyosin and the troponin complex, which are regulatory proteins, coordinate with each other to switch the myosin activity [107]. Connectin [108] (titin [109]), nebulin [110], tropomodulin [111], α-actinin [68], Fhod3 [112] and others, which are structural proteins, are involved in myofibril formation and maintenance, elasticity and elongation [113].

The contractile ring is a well-known actin-based machinery with an important function in eukaryotic cell division [114]. The contractile ring is also composed of contractile proteins, actin filaments and myosin II, and associated proteins such as formin [115], anillin [116] and septin [117]. α-Actinin and fimbrin cooperate with myosin II when the contractile ring is organized [118].

Muscles and contractile rings are categorized as actomyosin-based machinery, along with stress fibers, because their power source is actomyosin-based motors. We suspect that in these machineries, ABP-induced cooperative structural changes in actin filaments also cause allosteric regulatory effects on other ABP interactions. Further research is needed to uncover the relationships between these allosteric regulatory mechanisms.

Elongation of invadopodia of cancer cells requires microtubules and vimentin intermediate filaments in addition to filopodia machinery [119]. We recently discovered that microtubule elongation along actin filaments induced by microtubule-associated protein-4 contributes to the formation of cellular protrusions [120]. These results suggest the possibility that cancer invasion is regulated by complex and dynamic cell functions expressed by the cooperative interaction between the actin-based machinery and the microtubule-based machinery.

## 4. Significance of Propagation Distance and Directionality of Long-Range Allostery

Cofilin binds to actin filaments and induces structural changes accompanied by supertwisting in a relatively narrow range of around 10 actin protomers [32,39]. As a result, dense cofilin clusters are formed in a narrow section of the actin filament, and filaments that accumulate strain are severed at the end or inside of the cluster [32]. Such a narrow range of transmission of conformational change presumably results in dense clustering of cofilin, leading to severing of actin filaments by the cluster. In a moderate range of about 50 actin protomers, transmission of conformational changes induced by myosin II [11], tropomyosin [25] and α-actinin [47] may be involved in the formation and stabilization of actin-based machineries such as stress fibers, muscles and contractile rings. Finally, there are cases of structural changes over a much longer distance. For example, more than 160 actin protomers are affected by formin [53], and this would be useful to prevent cofilin binding to actin filaments in a wide, anterior area of lamellipodia.

Directionality of the propagation of structural changes may also play an important role in functional expression of actin-based machineries. For example, the unidirectionality of the cooperative interaction of myosin II with actin filaments [23] presumably plays a role in the unidirectional motor properties of myosin. The mechanism of the cooperative structural change of actin filaments is not well understood, but unidirectional propagation of structural change may be an inherent property of polar cables. In other words, unidirectional propagation of cooperative structural changes may be a consequence rather than being of physiological significance.

As described above, the information transmitted by a structural change in actin filament cables may be important for the formation and function of the actin-based machinery in cells (Figure 3). In addition to the stress fibers and lamellipodia discussed in this review, actin filaments and various ABPs interact with each other in filopodia and the cell cortex, and long-range allostery may play important roles in the expression of their functions. In the future, studies focusing on the types, directions, and propagation distances of cooperative actin filament structural changes induced by the binding of ABPs will elucidate how actin-based machineries are assembled and function.

Similar long-range allostery has been discovered in microtubules as well. Muto et al. reported that kinesin-1 binding to microtubules in the presence of ATP causes a long-range structural change of microtubules, increasing their affinity for kinesin toward the plus end [121]. Shima et al. reported that kinesin-1 binding induces conformation switching of microtubules to increase the affinity for kinesin-1 [122]. They concluded that the biased transport triggered by this positive feedback loop would specify a future axon [122].

## 5. Future Perspectives

Future studies are needed to reveal detailed allosteric interactions between actin filaments and ABPs whose cooperative interactions with actin filaments are not well analyzed, including drebrin, fimbrin, filamin and gelsolin. Those studies will provide important information to unveil the entire picture of how various ABPs control the formation and function of actin-based machineries through long-range allosteric regulation driven by cooperative conformational changes of actin filaments.

Mutation of the Lys-336 residues of α-actin, which is located near the ATP binding site, to Ile or Asp causes congenital myopathy [123,124]. Umeki et al. reported that K336I actin forms apparently normal cofilaments with wild type actin, but the interactions between the cofilaments and α-actinin, cofilin, and myosin II are impaired in vitro [41]. Noguchi et al. also revealed that the G146V mutation, which causes dominant lethality in yeast, cooperatively inhibits cofilin binding in vitro [39]. These results support the idea that ABP-dependent cooperative structural changes in actin protomers, or the long-range allostery, play important roles in essential cellular activities and that the defect in the long-range allostery can cause disease. Clearly, more work is needed to understand the relationship between diseases and actin’s long-range allostery.

## 6. Conclusions

Actin filaments express some types of long-range allostery caused by an interaction with ABPs, which have different propagation distances of cooperative structural change. These cooperative structural changes influence each other, so that actin-based machineries can be properly formed and function. Research to elucidate the function of the actin-based machinery through the cooperative structural change of actin filaments is still in its infancy. This long-range allostery, found not only in actin filaments but also in microtubules, will be the key to understanding differences between the biological and artificial machineries. That is, artificial machines composed of gears, motors, etc., are designed without considering the flexibility and cooperativity of the parts. In contrast, in biological machineries, the functions realized by the high degree of cooperativity of parts themselves play important roles in the operation mechanism. Protein molecules are by no means simply smaller versions of parts of artificial machines.

## Figures and Tables

**Figure 1 ijms-21-03209-f001:**
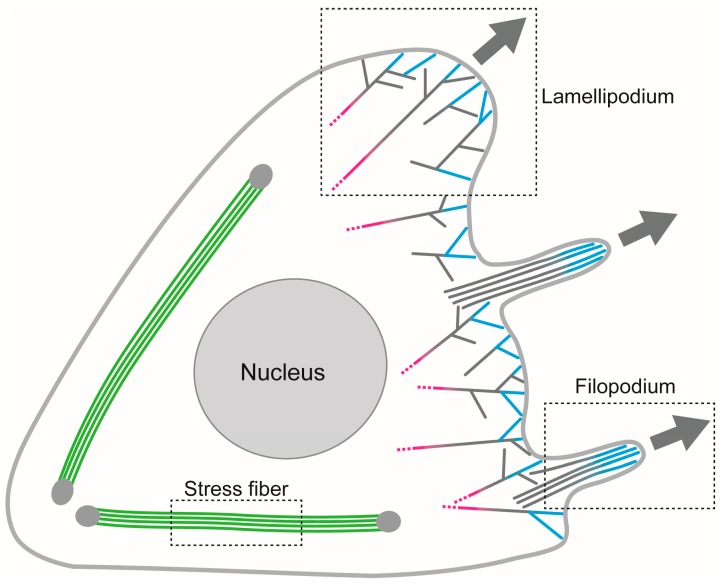
Possible contribution of allosteric interaction between actin-binding proteins (ABPs) and actin filaments to the formation and functions of actin-based machineries, including stress fiber, lamellipodium, and filopodium. Green filaments indicate actin filaments along which cooperative structural changes are evoked by myosin II. Blue filaments in lamellipodium and filopodia are polymerized by formin bound to the barbed end near the cell membrane [78]. Formin would induce a cooperative structural change, but those structural changes do not reach all the way to the pointed ends of filaments (gray sections of the filaments). The binding of cofilin to the gray regions induced another type of structural change (magenta filaments), so that the actin filaments are depolymerized in regions deeper inside the cell. The detailed mechanism driven by cooperative structural changes in actin filaments constituting the stress fibers, lamellipodia and filopodia will be explained in the following sections.

**Figure 2 ijms-21-03209-f002:**
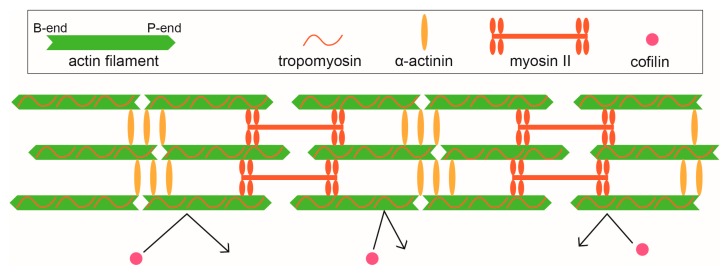
Stress fiber composed of tropomyosin, α-actinin, and myosin II, and the proposed mechanism for how cofilin is excluded. The binding of myosin II (and probably tropomyosin and α-actinin) to actin filaments induces a cooperative structural change of filaments accompanied by untwisting of the helix (green filaments). In addition, stress fibers are also tensed by the movement of myosin II. The untwisting and tension of the actin filaments presumably involve constructing a stress fiber that repels cofilin.

**Figure 3 ijms-21-03209-f003:**
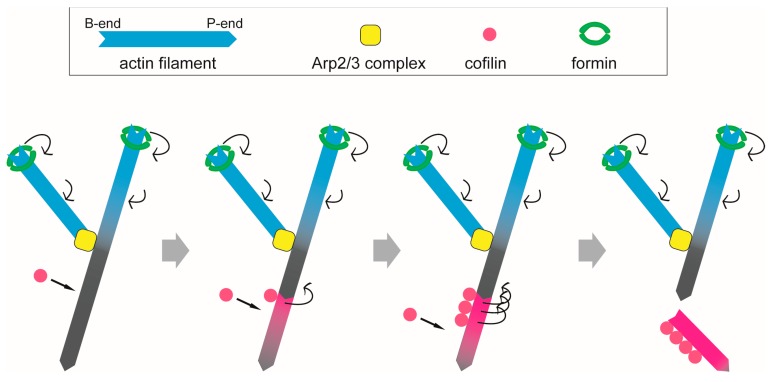
Severing of formin-bound actin filament by cofilin in lamellipodia. Formin bound to the barbed end of an actin filament induces untwisting of the filament (blue), but the effect does not reach the pointed end of filaments (grey) deeper inside the cells because the untwisting is presumably blocked at the branching point where ARP2/3 complex is connected. Cofilin binds to the grey section of the filament unaffected by formin and causes cooperative structural changes with supertwisting of filaments (magenta). Cofilin clusters grow in the magenta area, and the actin filament is severed.

**Figure 4 ijms-21-03209-f004:**
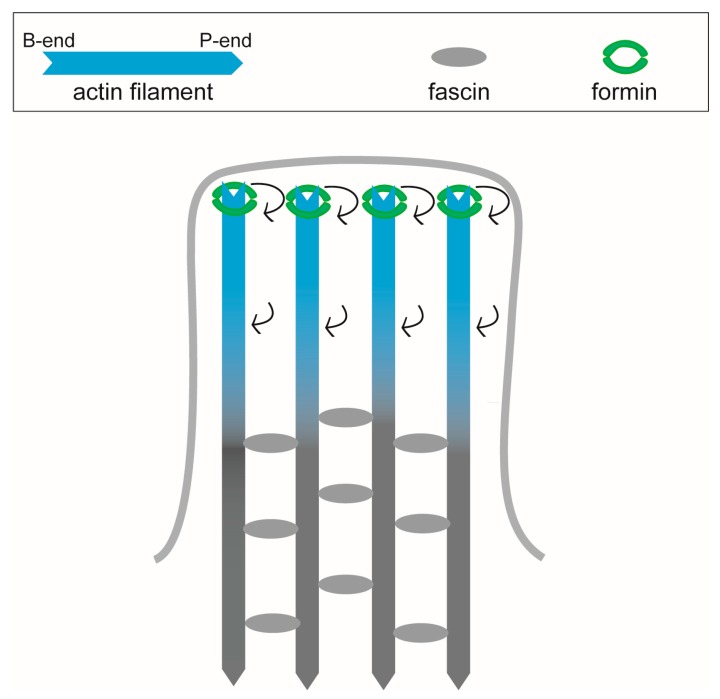
Extension and stabilization of filopodia. Similar to lamellipodia, in filopodia, formin binds to the barbed end of actin filaments and untwists the filaments. Since fascin can bind to actin filaments regardless of this structural change, filaments are bundled in filopodia. The formin-induced untwisting is presumably blocked and not propagated to the pointed-end of filaments by the immobilization by fascin. Cofilin cannot bind not only to the untwisted segments near the tip of filopodia, but also to the bundled sections, since those bundled sections cannot be supertwisted.

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
