# Peer review of "Long-Range and Directional Allostery of Actin Filaments Plays Important Roles in Various Cellular Activities"

_ijms, 2020, doi:10.3390/ijms21093209_

Round 1
Reviewer 1 Report
In this review by Tokuraku, et al., the authors present a brief summary of select actin-binding proteins (ABPs) and how these ABPs bind to actin filaments in various regions of the cytoplasm to induce conformational changes of the actin filaments. The authors’ first part of the review discusses various ABPs, including those that have three binding “styles”. Next, they discuss the role of these ABPs in modification of actin filaments in various cellular structures, including stress fibers and lamellipodia. In closing, the authors very briefly discuss how the binding of these ABPs can participate in regulation of the actin cytoskeleton to induce physiological changes. This review provides a brief, systematic summary of a select number of ABPs and how they regulate the dynamic nature of actin filaments. As presently written, this review currently leaves out a few major actin-containing cellular structures, particularly in a disease context. Addition of this information would greatly strengthen this review, in addition to a slight revision of some of the figures and the review’s title. The following changes should be addressed by the authors prior to acceptance of this review for publication.
- The title of the review is very heavily centered on describing the actin filaments as “information transmission cables”. However, the review’s text focuses very little on the physiological consequences of these allosteric changes imposed by these ABPs mentioned. The review focuses more on the physical changes to the actin filaments. It would be beneficial to discuss the physiological consequences of these physical changes both in normal and aberrant contexts, such as in cancer and other diseases.
- Also related to the current title of the review, which mentions “narrow intracellular spaces”, but only mentions lamellipodia within the current text. There are many other narrow intracellular spaces within the cell that are actin-rich, including filopodia (which is only briefly mentioned once in the text) and invadopodia, that are not discussed in this review. Discussing these structures, which have very dynamic actin activity, would greatly benefit the scope of this review and broaden the audience.
- In Figure 3, the color and shape used to depict cofilin (a magenta line at the end of the lamellipodia) is too difficult to see. The authors should use a different shape to represent cofilin (as they do in figure 2) to make it easier for the reader to see.
Author Response
"The title of the review is very heavily centered on describing the actin filaments as “information transmission cables”. However, the review’s text focuses very little on the physiological consequences of these allosteric changes imposed by these ABPs mentioned. The review focuses more on the physical changes to the actin filaments."
We have revised the title and abstract to match the content of the text, according to the opinion of reviewers # 1 and # 3.
"It would be beneficial to discuss the physiological consequences of these physical changes both in normal and aberrant contexts, such as in cancer and other diseases."
According to the reviewers' suggestion, we have added descriptions about the disease (Lines 343–348 and Lines 389–397).
"Also related to the current title of the review, which mentions “narrow intracellular spaces”, but only mentions lamellipodia within the current text. There are many other narrow intracellular spaces within the cell that are actin-rich, including filopodia (which is only briefly mentioned once in the text) and invadopodia, that are not discussed in this review. Discussing these structures, which have very dynamic actin activity, would greatly benefit the scope of this review and broaden the audience."
Following the reviewer's suggestion, we have added new section 3.3. about filopoidia (Line 298–323). Moreover, a new paragraph is added regarding invadopodia, and discussed in the context of cross-talk with microtubule-based machineries (Line 343–348).
"In Figure 3, the color and shape used to depict cofilin (a magenta line at the end of the lamellipodia) is too difficult to see. The authors should use a different shape to represent cofilin (as they do in figure 2) to make it easier for the reader to see."
We have modified Fig. 3 (Fig. 1 in the new manuscript) and the legend accroding to reviewers' comment.
Reviewer 2 Report
In this review article entitled “Role of actin filaments as information transmission cables through long-range and directional allostery in narrow intracellular spaces”, Tokuraku et al. provide a general overview on what is known about long-range allosteric regulation driven by cooperative conformational changes of actin filaments that are evoked by binding of ABPs. The importance of this topic has become evident over the last years and a review focusing on this subject is timely and relevant. The review starts with a brief introduction followed by a section entitled “Cooperative binding of ABPs”, where the authors describe various ABPs that bind to and induce conformational changes in actin filaments. Then, the authors describe the importance of allosteric interaction between some of the ABPs and actin filaments, focusing in stress fibers and lamellipodia. Finally, the authors mention about the significance of such allosteric interaction. My comments on the manuscript are listed below.
1- Figures 1 and 2 were not cited in the main text. Figure 3 was the only one cited. Please, cite the figures throughout the text so that the readers could analyze them while they are reading.
2- Still regarding the figures, I suggest modifying their order of appearance. Figure 3 should be new Figure 1. Figures 1 and 2 should be the new Figures 2 and 3, respectively. In addition, the authors should emphasize somehow that Figures 2 and 3 are detailed representations of what could happen in specific regions of figure 1 (stress fibers and lamellipodia).
3- Moreover, the new Figures 2 and 3 (old Figures 1 and 2) should be more representative of what is described in the text. For example, the authors should represent the locations of the different tropomyosins mentioned. Also, the authors said that cofilin is able to supertwist the actin filament and binding of myosin II, untwists and generates tension on it. However, none of these mechanical descriptions were represented in the old figure 1 (new figure 2). The same for old figure 2 (new Figure 3). Although the authors described that formin decreases the twist and cofilin supertwists the actin filament, they tried to represent it with a color code. However, I suggest changing this representation to something more visually understandable. The present figures do not talk to the manuscript text or even represent the interesting phenomena described in the review.
4- The authors should create a new section, related to future perspectives. In this section the authors should conjecture possible long-range allosteric regulation driven by cooperative conformational changes of actin filaments that are evoked by some of the ABPs that were mentioned (Drebrin, Fimbrin, Filamin, Geosolin) but not described so far.
Some other specific comments:
- Line 11: “carry out” should be modified by “are implicated” or “are involved”;
- Line 41: the phrase “but not in direct contact with the initial ABP in vitro” should be better explained;
- Line 143: please, insert a space between the words “of” and “alfa-actinin”;
- Line 204: “This would explain” should be “This would partially explain”;
- Line 222: please, change the word “macrophage” to “macrophages”;
- Line 222: please, change “neural crest cells” to “neural crest derived cells”;
- Line 305: please, the authors should emphasize somehow, that the 3 long-range allosteric interactions (<14 protomers, ~50 protomers, >100 protomers) are the ones described so far. But there may be others.
Author Response
"1- Figures 1 and 2 were not cited in the main text. Figure 3 was the only one cited. Please, cite the figures throughout the text so that the readers could analyze them while they are reading. "
According to the reviewers' instruction, we cited all figures in the text.
"2- Still regarding the figures, I suggest modifying their order of appearance. Figure 3 should be new Figure 1. Figures 1 and 2 should be the new Figures 2 and 3, respectively. In addition, the authors should emphasize somehow that Figures 2 and 3 are detailed representations of what could happen in specific regions of figure 1 (stress fibers and lamellipodia)."
The order of the figures was changed as instructed by the reviewer, and stress fibers, lamellipodia, and filopodia were specified in the new Fig. 1.
"3- Moreover, the new Figures 2 and 3 (old Figures 1 and 2) should be more representative of what is described in the text. For example, the authors should represent the locations of the different tropomyosins mentioned. Also, the authors said that cofilin is able to supertwist the actin filament and binding of myosin II, untwists and generates tension on it. However, none of these mechanical descriptions were represented in the old figure 1 (new figure 2). The same for old figure 2 (new Figure 3). Although the authors described that formin decreases the twist and cofilin supertwists the actin filament, they tried to represent it with a color code. However, I suggest changing this representation to something more visually understandable. The present figures do not talk to the manuscript text or even represent the interesting phenomena described in the review."
We added detailed descriptions in the text, and modified the figures and figure legends.
"4- The authors should create a new section, related to future perspectives. In this section the authors should conjecture possible long-range allosteric regulation driven by cooperative conformational changes of actin filaments that are evoked by some of the ABPs that were mentioned (Drebrin, Fimbrin, Filamin, Geosolin) but not described so far."
We added a new section (5) related to the future perspectives (Lines 383–397).
"Some other specific comments:
Line 11: “carry out” should be modified by “are implicated” or “are involved”;"
Following the reviewer's instruction, we modified "carry out" to "are involved" (Line 12).
"Line 41: the phrase “but not in direct contact with the initial ABP in vitro” should be better explained;"
We corrected the explanation (Line 45).
"Line 143: please, insert a space between the words “of” and “alfa-actinin”;"
We inserted a space between "of" and "alfa-actinin" (Line 149).
"Line 204: “This would explain” should be “This would partially explain”;"
We added "partially" as suggested by the reviewer (Line 225).
"Line 222: please, change the word “macrophage” to “macrophages”;"
We changed the word "macrophage" to “macrophages" (Line 265).
"Line 222: please, change “neural crest cells” to “neural crest derived cells”;"
We changed "neural crest cells" to "neural crest derived cells" (Line 265).
"Line 305: please, the authors should emphasize somehow, that the 3 long-range allosteric interactions (<14 protomers, ~50 protomers, >100 protomers) are the ones described so far. But there may be others."
We remove this description because it does not affect the argument (Line 400).
Reviewer 3 Report
I would recommend some amendments to the title of the review:
Most of the review talks about long-range, directional allostery and how
these interactions are important to for various actin-based structures,
but the idea of information transmission cables comes in at the end and
is not central to the review. The authors should either describe in the
review how this information transmission is crucial to the structures
formed more clearly or rewrite the title to emphasize more on the
general functions of this allostery
The title also mentions narrow intracellular spaces. However, not all of
the examples the authors provide are of narrow intracellular spaces
(stress fibers are present in the cell body, lamellopodia are thin but
spread out). Generally one would think of filopodia, villi, dendrites
etc to be narrow intracellular spaces.
‘Information transmission’ needs to be described more clearly in the abstract – what information is being transmitted? What is the function
of this information transmission?
2.1.1 mentions myosin II and V – are these the only two myosins known to
induce long-range allostery? Even if it is not known for the rest of the
myosins, it would be good if the authors can mention what they think is
the case with other myosins, taking into account the specific structural
similarities and differences between various members of the myosin family.
2.2.2 – it appears from the text here that there isn’t conclusive
evidence for long-range allostery by alpha-actinin. There is evidence of
co-operativity, but not for conformational changes in actin upon
alpha-actinin binding. If this is true, it should be emphasized that
while it is likely to be the case, the evidence for long-range allostery
remains to be shown.
2.2.3 – Similar to 2.2.2 it appears that there is no direct evidence of
long-range allostery induced by filamin but it is likely that is does so
based on indirect evidence. Again, this should be stated clearly.
In the introduction for section 3, I would recommend mentioning other
actin-based structures in specialized cells such as microvilli, cilia,
stereocilia etc to demonstrate how diverse actin morphology can be and
how important it is for specific ABPs to be recruited to these
structures for proper functioning of a diverse range of processes from
hearing to motility.
The lines 310-313 in the conclusion about artificial machineries are not
very clear. It is not obvious how long-range allostery will help
understand differences between biological and artificial machineries and
what the ‘different perspective’ is. Please clarify or remove this
sentence.
Author Response
"I would recommend some amendments to the title of the review:
Most of the review talks about long-range, directional allostery and how these interactions are important to for various actin-based structures, but the idea of information transmission cables comes in at the end and is not central to the review. The authors should either describe in the review how this information transmission is crucial to the structures formed more clearly or rewrite the title to emphasize more on the general functions of this allostery"
We have revised the title and abstract to match the content of the review, according to the opinions of reviewers # 1 and # 3.
"The title also mentions narrow intracellular spaces. However, not all of the examples the authors provide are of narrow intracellular spaces (stress fibers are present in the cell body, lamellopodia are thin but spread out). Generally one would think of filopodia, villi, dendrites etc to be narrow intracellular spaces."
According to the reviewer's suggestion, we have changed the title, and added a section dealing with filopodia (Line298-323) and a paragraph dealing with invadopodia (Lines 343–348).
"‘Information transmission’ needs to be described more clearly in the abstract – what information is being transmitted? What is the function of this information transmission?"
The expression in the abstract was changed, and the word “information transmission” is removed (Lines 19–22).
"2.1.1 mentions myosin II and V – are these the only two myosins known to induce long-range allostery? Even if it is not known for the rest of the myosins, it would be good if the authors can mention what they think is the case with other myosins, taking into account the specific structural similarities and differences between various members of the myosin family."
In this paper, we focused on myosin II and V, which are well known for their cooperative binding to actin filaments. So we changed the subtitle of 2.1.1. to myosin II and V to avoid confusion with other myosins. We also added a description of myosin II isoforms according to the reviewers' suggestion (Lines 236–253).
"2.2.2 – it appears from the text here that there isn’t conclusive evidence for long-range allostery by alpha-actinin. There is evidence of co-operativity, but not for conformational changes in actin upon alpha-actinin binding. If this is true, it should be emphasized that while it is likely to be the case, the evidence for long-range allostery remains to be shown.
2.2.3 – Similar to 2.2.2 it appears that there is no direct evidence of long-range allostery induced by filamin but it is likely that is does so based on indirect evidence. Again, this should be stated clearly."
As the reviewer pointed out, there is no direct evidence of cooperative structural changes of actin filaments that are induced by α-actinin and filamin. Therefore, we added a description that further research is needed on the long-range allosteries of α-actinin and filamin (Lines 159–160).
"In the introduction for section 3, I would recommend mentioning other actin-based structures in specialized cells such as microvilli, cilia, stereocilia etc to demonstrate how diverse actin morphology can be and how important it is for specific ABPs to be recruited to these structures for proper functioning of a diverse range of processes from hearing to motility."
We added the relevant description according to the reviewers' suggestion (Lines 35–38).
"The lines 310-313 in the conclusion about artificial machineries are not very clear. It is not obvious how long-range allostery will help understand differences between biological and artificial machineries and what the ‘different perspective’ is. Please clarify or remove this sentence."
According to reviewers' suggestion, we added explanations (Lines 406–410).
Round 2
Reviewer 1 Report
The authors have satisfied my original concerns surrounding the manuscript. I now recommend that it be accepted for publication in its current form.
Reviewer 2 Report
The authors performed significant changes that answered all my comments, suggestions and concerns. I have no further questions.